

# Genome-wide analysis of circular RNAs in goat skin fibroblast cells in response to Orf virus infection

Feng Pang, Mengmeng Zhang, Xiaojian Yang, Guohua Li, Shu Zhu, Xin Nie, Ruiyong Cao, Xiaohong Yang, Zhenxing Zhang, Haifeng Huang, Baobao Li, Chengqiang Wang, Li Du and Fengyang Wang

College of Animal Science and Technology, College of Tropical Agriculture and Forestry, Hainan University, Hainan Key Lab of Tropical Animal Reproduction & Breeding and Epidemic Disease Research, Hainan University, Haikou, China

Corresponding author
Fengyang Wang, fywang68@163.com

## ABSTRACT

Orf, caused by Orf virus (ORFV), is a globally distributed zoonotic disease responsible for serious economic losses in the agricultural sector. However, the mechanism underlying ORFV infection remains largely unknown. Circular RNAs (circRNAs), a novel type of endogenous non-coding RNAs, play important roles in various pathological processes but their involvement in ORFV infection and host response is unclear. In the current study, whole transcriptome sequencing and small RNA sequencing were performed in ORFV-infected goat skin fibroblast cells and uninfected cells. A total of 151 circRNAs, 341 messenger RNAs (mRNAs), and 56 microRNAs (miRNAs) were differently expressed following ORFV infection. Four circRNAs: circRNA1001, circRNA1684, circRNA3127 and circRNA7880 were validated by qRT-PCR and Sanger sequencing. Gene ontology (GO) analysis indicated that host genes of differently expressed circRNAs were significantly enriched in regulation of inflammatory response, epithelial structure maintenance, positive regulation of cell migration, positive regulation of ubiquitin-protein transferase activity, regulation of ion transmembrane transport, etc. The constructed circRNA-miRNA-mRNA network suggested that circRNAs may function as miRNA sponges indirectly regulating gene expression following ORFV infection. Our study presented the first comprehensive profiles of circRNAs in response to ORFV infection, thus providing new clues for the mechanisms of interactions between ORFV and the host.

# INTRODUCTION

Circular RNAs (circRNAs), a new member of non-coding RNAs, are generally produced by back-splicing of pre-mRNA(*Barrett & Salzman, 2016*; *Barrett, Wang & Salzman, 2015*). Because of their unique circular structure, circRNAs can resist the activity of RNA exonuclease digestion. Thus, they are more stable and have longer half-lives than their linear transcripts *in vivo* (*Jeck et al., 2013*; *Suzuki et al., 2006*). Moreover, circRNAs are highly abundant and are conserved among a variety of species (*Barrett & Salzman, 2016*;

*Memczak et al., 2013*). A number of circRNAs also present cell type- and tissue-specific expression patterns (*Salzman et al., 2013*; *Szabo et al., 2015*).

Increasing evidence suggests that circRNAs can function as competing endogenous RNAs (ceRNAs) or miRNA sponges, thereby acting as post-transcriptional regulatory elements (*Hansen et al., 2013*; *Qu et al., 2015*; *Zhang et al., 2016*). For instance, the circRNA ciRS-7, containing 73 conventional miR-7 binding sites, strongly inhibits miR-7 activity, thus increasing the expression levels of miR-7 targets (*Hansen et al., 2013*). Many studies show that circRNAs play key roles in the progression of several serious types of disease such as cancer (*Li et al., 2015*; *He et al., 2017a*; *Tian et al., 2017*). Moreover, circRNAs are associated with viral infection and host-pathogen interactions. Shi et al. performed circRNA sequencing to study the expression profiles of circRNAs in Vero cells following infection with SV40 virus (*Shi et al., 2017*) while *He et al. (2017b)* analyzed the roles of circRNA in host-grass carp reovirus interactions via deep sequencing. However, the functions of circRNAs within host cells in response to Orf virus (ORFV) infection has not been studied thus far.

Orf, also called contagious ecthyma, is a globally-distributed zoonotic disease responsible for serious economic losses in the agricultural and animal husbandry industries (*Friederichs et al., 2014*). Infection usually occurs in the lips, oral mucosa, and around the nostrils of goats and sheep, with typical symptoms including erythema, papula, pustules, and scabs(*Delhon et al., 2004*). ORFV, the causative agent of Orf, is a zoonotic virus belonging to the *Parapoxvirus* genus (*Zhang et al., 2014a*). As such, humans may become infected with ORFV after coming into contact with infected animals (*Kumar et al., 2014*; *Peng et al., 2016*; *Rajkomar et al., 2016*). ORFV has a linear, double-stranded DNA genome approximately 130–140 kb in length and containing a putative 131 open reading frames (*Delhon et al., 2004*; *Martins et al., 2017*; *Zhang et al., 2014a*). To better understand the pathogenesis of ORFV, several previous studies have conducted genome sequencing of different ORFV isolates (*Chi et al., 2015*; *Delhon et al., 2004*; *Zhao et al., 2010*), as well as transcriptome sequencing of host cells following ORFV infection (*Chen et al., 2017*; *Jia et al., 2017*).

However, the specific characteristics and roles of circRNAs during ORFV infection remain unclear. Therefore, in the current study we conducted circRNA sequencing and small RNA sequencing of uninfected and ORFV-infected goat skin fibroblast (GSF) cells in an attempt to gain new insights into ORFV pathogenesis.

## MATERIAL AND METHODS

### Cell culture and viral infection

Goat skin fibroblast cells (GSF cells) were obtained from the Cell Bank of the Chinese Academy of Science (Kunming, China) and maintained in Dulbecco's modified Eagle medium (DMEM; Invitrogen, Carlsbad, CA, USA) supplemented with 10% fetal bovine serum (Invitrogen) at 37 °C and 5% $CO_2$. When GSF cells seeded in 10-cm diameter dish reached ~90% confluence, the medium was removed and cells were rinsed three times with PBS. Cells were then infected with ORFV strain JLSY (tissue culture infectious dose 50% = $10^{6.2}$/mL), a gift from Professor Guixue Hu and Research Associate Hongze Shao, at a

multiplicity of infection of one. Following incubation for 1 h at 37 °C, the virus suspension was removed and cells were cultured for a further 6 h in standard medium.

## RNA extraction, library construction, and sequencing

Total RNA from both ORFV infected GSF samples (OV) and uninfected GSF samples were isolated using an Ambion *mir* Vana miRNA Isolation Kit (Thermo Fisher Scientific, Waltham, MA, USA). The quality of total RNA were analyzed by Bioanalyzer 2100 (Agilent Technologies, Santa Clara, CA, USA), and the concentration of the total RNA was quantified using a NanoDrop 2000 (Thermo Fisher Scientific, Lafayette, CO, USA). In this study, circRNA libraries were constructed as below: Ribosomal RNA was removed from 5 μg aliquots of total RNA using an Epicentre Ribo-Zero Gold Kit (Illumina, San Diego, CA, USA). Then, the RNA fractions were fragmented and were reverse transcribed using an mRNA-Seq Sample Preparation Kit (Illumina, San Diego, CA, USA). The cDNA libraries were then paired-end sequenced using an Illumina HiSeq 4000 platform (Lc-bio, Hangzhou, China). The raw and processed data have been deposited into the Gene Expression Omnibus database (https://www.ncbi.nlm.nih.gov/geo/) under accession number GSE121725. (2) Small RNA libraries: About 1 μg total RNA of each sample was used for cDNA library construction with the TruSeq Small RNA Preparation Kit (Illumina, San Diego, CA, USA) following the manufacturer's protocol. Then, the cDNA libraries were single-end 50 bp (SE50) sequenced with an Illumina HiSeq 2500 platform (Lc-bio, Hangzhou, China). The raw and processed data have been deposited into the Gene Expression Omnibus database (https://www.ncbi.nlm.nih.gov/geo/) under accession number GSE121726.

## Identification and differential expression analysis of circRNAs and mRNAs, and miRNAs

Firstly, Cutadapt (*Martin, 2011*) was utilized to remove reads containing undetermined bases, adaptors, and low quality bases. Then sequence quality was verified using FastQC (http://www.bioinformatics.babraham.ac.uk/projects/fastqc/). Bowtie2 (*Langmead & Salzberg, 2012*) and Tophat2 (*Kim et al., 2013*) were used to map reads to the *Capra hircus* (goat) reference genome (RefSeq assembly accession: GCF_001704415.1). The matched reads of each sample were assembled using StringTie (*Pertea et al., 2015*). StringTie and Ballgown (*Frazee et al., 2015*) were utilized to evaluate the expression levels of all transcripts by Fragments per kilobase per million reads (FPKM). The dysregulated mRNAs were selected with $|\log_2 (\text{fold change})| \geq 1$ and $p \leq 0.05$ by R package Ballgown (*Frazee et al., 2015*).

Any unmapped reads were individually mapped to the goat reference genome by TopHat-Fusion (*Kim & Salzberg, 2011*). Then, reads mapped to the goat genome using TopHat-Fusion were analyzed by CIRCexplorer to identify candidate circRNAs (*Zhang et al., 2016*; *Zhang et al., 2014b*). The criteria were as follows: (1) GU/AG must occur at both ends of splice sites; (2) less than two mismatches; (3) more than one back-spliced junction read in at least one sample of GSF or OV group; (4) two splice sites are no more than 100 kb apart on the genome. The expression of circRNAs was calculated by the number of reads spanning back-splicing junction and FPKM was used to normalize the expression

level of circRNAs. CircRNAs with $|\log_2 \text{(fold change)}| \geq 1$ and $p \leq 0.05$ were regarded as differentially expressed by R package-edgeR (*Robinson, Mccarthy & Smyth, 2014*).

For miRNA analysis, ACGT101-miR (LC Sciences, Houston, Texas, USA) was used to acquire clean reads. Then, unique sequences containing 18 to 26 nucleotides were mapped to miRBase 21.0 by BLAST search to identify known and novel miRNAs. The expression level of miRNAs was normalized based on the read counts to tags per million counts (TPM). The significance standard was $p \leq 0.05$.

## Gene ontology (GO) and Kyoto Encyclopedia of Genes and Genomes (KEGG) analysis

GO analysis was used to determine significantly enriched GO terms ($p \leq 0.05$) by hypergeometric test (*Beißbarth & Speed, 2004*; *Gene Ontology Consortium, 2004*). KEGG pathway analysis was used to explore significantly enriched pathways ($p \leq 0.05$) by hypergeometric test (*Kanehisa et al., 2017*; *Kanehisa et al., 2012*).

$$P = 1 - \sum_{i=0}^{m-1} \frac{\binom{M}{i}\binom{N-M}{m-i}}{\binom{N}{n}}$$

$N$, Total number of circRNA-hosting genes; $n$, The number of circRNA-hosting genes with differential expression; $M$, The number of circRNA-hosting genes annotated to the GO term; $m$, The number of circRNA-hosting genes with differential expression annotated to the GO term.

## Bioinformatics analysis and ceRNA network construction

The circRNA-miRNA and miRNA-mRNA interactions were predicted using TargetScan 7.0 and miRanda software. TargetScan 7.0 predicts the targets of miRNAs based on seed region homologies (*Agarwal et al., 2015*) while MiRanda is mainly based on the combination of free energy generated by miRNAs binding to their target genes (*Betel et al., 2008*). The lower the free energy, the stronger the binding. TargetScan score percentiles $\geq 50$, Miranda max free energy values $< -10$ and Miranda score $>140$ were defined as the cutoff points for targets predicti. To further explore the functional role of circRNAs, a ceRNA network was constructed using Cytoscape 3.6.0 software (*Shannon et al., 2003*).

## Validation of miRNAs

Six differentially expressed novel miRNAs (PC-3p-8215_174, PC-5p-406_14064, PC-5p-2253_1210, PC-5p-5127_361, PC-3p-10316_124 and PC-3p-4306_468) with mean TPM $>30$ in GSF samples or OV samples were selected for qPCR validation. First, the total RNA was reverse- transcribed using a miRNA 1st Strand cDNA Synthesis kit (Vazyme, Nanjing, China) with specific stem-loop primers (Table S1). Then, qRT-PCR assays were performed using miRNA Universal SYBR qPCR Master Mix (Vazyme, Nanjing, China) with specific forward primers (F) and the universal reverse primer (Table S1) with an ABI 7500 Real-Time PCR System (Applied Biosystems, Foster City, CA, USA). U6 snRNA was used as an internal control for normalization of the expression level of these miRNAs. All experiments were conducted independently three times.

## Validation of circRNAs by qRT-PCR and Sanger sequencing

Differentially expressed circRNAs with mean FPKM $\geq$ 10 (relatively high expression ) in OV samples or GSF samples as well as possessing more than one back-spliced read in at least two replicates of OV samples or GSF samples were selected for confirmation by qRT-PCR. Thus, six down-regulated circRNAs: circRNA998, circRNA1000, circRNA1001, circRNA1684, circRNA3127, circRNA4287 and three up-regulated circRNAs: circRNA5112, circRNA7880, circRNA8565 were obtained. First, total RNA was reversely transcribed to cDNA using a Revert Aid First Strand cDNA Synthesis Kit (Thermo Fisher Scientific). Next, qPCR assays were performed with divergent primers using the ABI 7500 Real-Time PCR System (Applied Biosystems, Foster City, CA, USA) with $2\times$ SYBR qPCR Mix (Aidlab, Beijing, China). $2 - \Delta\Delta Ct$ method was used to calculate the relative expression level of circRNAs with goat glyceraldehyde-3-phosphate dehydrogenase (*GAPDH*) serving as an internal control. All experiments were conducted in triplicate. The PCR products from cDNA samples were ligated into pMD-19T (Takara, Dalian, China) for Sanger sequencing to determine the back-splicing junctions.

Moreover, genomic DNA(gDNA) was extracted from GSF cells using a Blood/Cell/Tissue Genome DNA Extraction Kit (Tiangen, Beijing, China). Both cDNA and gDNA were used as templates for PCR amplification using specific convergent and divergent primers (Table 1). The PCR products were examined using 1.5% agarose gel.

## Statistical analysis

Statistical significance analysis was performed by Student's $t$-test, and $p \leq 0.05$ was considered statistically significant.

## RESULTS

### Properties of circRNAs in ORFV-infected and uninfected GSF cells

We first performed circRNA sequencing, using the ribosomal RNA-depleted method, of three uninfected GSF samples (GSF samples) and three ORFV-infected samples (OV samples) on the Illumina HiSeq 4000 platform. We acquired an average of 86 million and 93 million raw reads for the GSF and OV groups, respectively. Clean reads, which accounted for >98.5% of the raw reads, were obtained after the removal of the low quality raw reads. The Q30 of each sample was ∼93%. Most reads (∼85%) were linearly mapped to the goat reference genome. Among the remaining unmapped reads, approximately 1% of reads from each sample were identified as back-spliced junction reads. Furthermore, we aligned the reads unmapped to goat genome from each sample to Orf virus reference genome (Orf virus NA1/11 strain under GenBank number KF234407.1). The results indicated that a total of 125,619 reads, 165,878 reads and 157,779 reads were mapped to Orf virus genome in OV-1, OV-2, and OV-3 sample, respectively. There were nearly zero reads aligned to Orf virus genome in uninfected GSF samples (Table 2).

Finally, 9,979 and 10,844 circRNAs with more than one back-spliced read existed in at least one sample of GSF or OV group were identified, respectively (Table S2). Among these circRNAs, 4,649 were shared between the groups, while 5,330 and 6,195 circRNAs were uniquely expressed in the GSF and OV samples, respectively (Fig. 1A).

**Table 1  List of convergent and divergent primers used for circRNAs qRT-PCR validation.**

| Primers | Sequences (5′–3′) | Size (bp) |
|---|---|---|
| GAPDH CON-F | AGCCGTAACTTCTGTGCTGT | 234 |
| GAPDH CON-R | TTCCCGTTCTCTGCCTTGAC | |
| GAPDH DIV-F | ATGGTCCACATGGCCTCC | 334 |
| GAPDH DIV-R | CATCTTGTCTCAGGGATGC | |
| circRNA1001 CON-F | CCACTCAGTTCCCTGCTGAT | 82 |
| circRNA1001 CON-R | TCTTTACTTTGTGGCTGGCTC | |
| circRNA1001 DIV-F | TAGACAGCTCTGACAGCATGG | 101 |
| circRNA1001 DIV-R | TGCCACATGACTCATTAATTTC | |
| circRNA1684 CON-F | GGAGTCAACCTCACCACTGA | 71 |
| circRNA1684 CON-R | CCCCGGTCATAGCACACAA | |
| circRNA1684 DIV-F | CGATTACTCCATGTACCAGGCA | 155 |
| circRNA1684 DIV-R | GCACACAAACCTGTAATCCTGG | |
| circRNA3127 CON-F | AGGACCCTCATCCCTCGTTA | 114 |
| circRNA3127 CON-R | GTCCACGGTGATGGATGAGTT | |
| circRNA3127 DIV-F | TCACCCTCAACTACCTCAGGCT | 78 |
| circRNA3127 DIV-R | GTCACCCCTCCTTCAAACACAG | |
| circRNA7880 CON-F | AAAAGAAGCCGTCTCGGACA | 132 |
| circRNA7880 CON-R | CCAGACGTTTTCTGGGGCTA | |
| circRNA7880 DIV-F | AATCAGATAGCCACCATCTTG | 85 |
| circRNA7880 DIV-R | TGTAGCCTGTGACTGGGAAC | |
| circRNA998 DIV-F | GACGACCTGATGGATTATCACC | 89 |
| circRNA998 DIV-R | TGCCATAATCTTGTTGGAATCA | |
| circRNA1000 DIV-F | TCGGAAACAACTGAACTTATGA | 125 |
| circRNA1000 DIV-R | TGTTCTTCACTTATACCCTCTGG | |
| circRNA4287 DIV-F | GTGTGAAAATAACGTGAAGGAA | 102 |
| circRNA4287 DIV-R | CTTCTAATTTCCTCACTCTCAGA | |
| circRNA5112 DIV-F | GGCTAAGCAATTCTCGGTTGG | 77 |
| circRNA5112 DIV-R | TTGTAGCCTGTGACTGGGAACG | |
| circRNA8565 DIV-F | GCTACTTCCAGCTGCAGATGTG | 137 |
| circRNA8565 DIV-R | ACACTGAGAACTTCAGGAACGC | |

Interestingly, approximately 98% of the circRNAs were exonic circRNAs (ecircRNAs), while the remaining 2% were circular intronic RNAs (ciRNA) based on their location in the goat genome (Fig. 1B). These candidate circRNAs were widely distributed from chromosome 1 to chromosome 29 (Fig. 1C). In particular, chromosomes 1, 2, 3, 10, and 11 produced more than 800 circRNAs, respectively. The lengths of the circRNAs varied from 300–1,300 bp (Fig. 1D). More than 2,000 genes produced only one circRNA isoform, while 889 genes produced two different isoforms, 554 genes produced three circRNA isoforms, and 364 genes produced four isoforms, with the number of genes decreasing with the number of isoforms produced (Fig. 1E).

Additionally, an average of 40,882 transcripts were detected in GSF samples while 40,778 transcripts were identified in three OV samples, respectively. Following miRNA sequencing, 695 mature miRNAs in GSF samples and 674 miRNAs in OV samples were acquired, respectively. The mapping overview of miRNA sequencing is provided in Table S3.

Pang et al. (2019), *PeerJ*, DOI 10.7717/peerj.6267

**Table 2  Summary of circRNA sequencing data.**

| Sample | GSF-1 | GSF-2 | GSF-3 | OV-1 | OV-2 | OV-3 |
|---|---|---|---|---|---|---|
| Raw reads | 84,146,788 | 89,058,270 | 85,890,172 | 89,408,452 | 93,833,004 | 97,376,392 |
| Valid reads | 81,678,856 | 86,169,216 | 83,244,598 | 86,920,674 | 91,594,088 | 95,061,748 |
| Mapped reads (linear) | 70466868(86.27%) | 72820127(84.51%) | 70690768(84.92%) | 75616664(87.00%) | 79582058(86.89%) | 82061025(86.32%) |
| Unmapped reads | 11211988(13.73%) | 13349089(15.49%) | 12553830(15.08%) | 11304010(13.00%) | 12012030(13.11%) | 13000723(13.68%) |
| Back-spliced junctions reads | 799766(0.98%) | 1133133(1.32%) | 905843(1.09%) | 874057(1.01%) | 766106(0.84%) | 889334(0.94%) |
| Reads mapped to Orf virus | 41(0.00%) | 28(0.00%) | 17(0.00%) | 125619(1.11%) | 165878(1.38%) | 157779(1.21%) |
| Q30 | 93.38% | 92.27% | 93.08% | 94.02% | 93.80% | 93.73% |

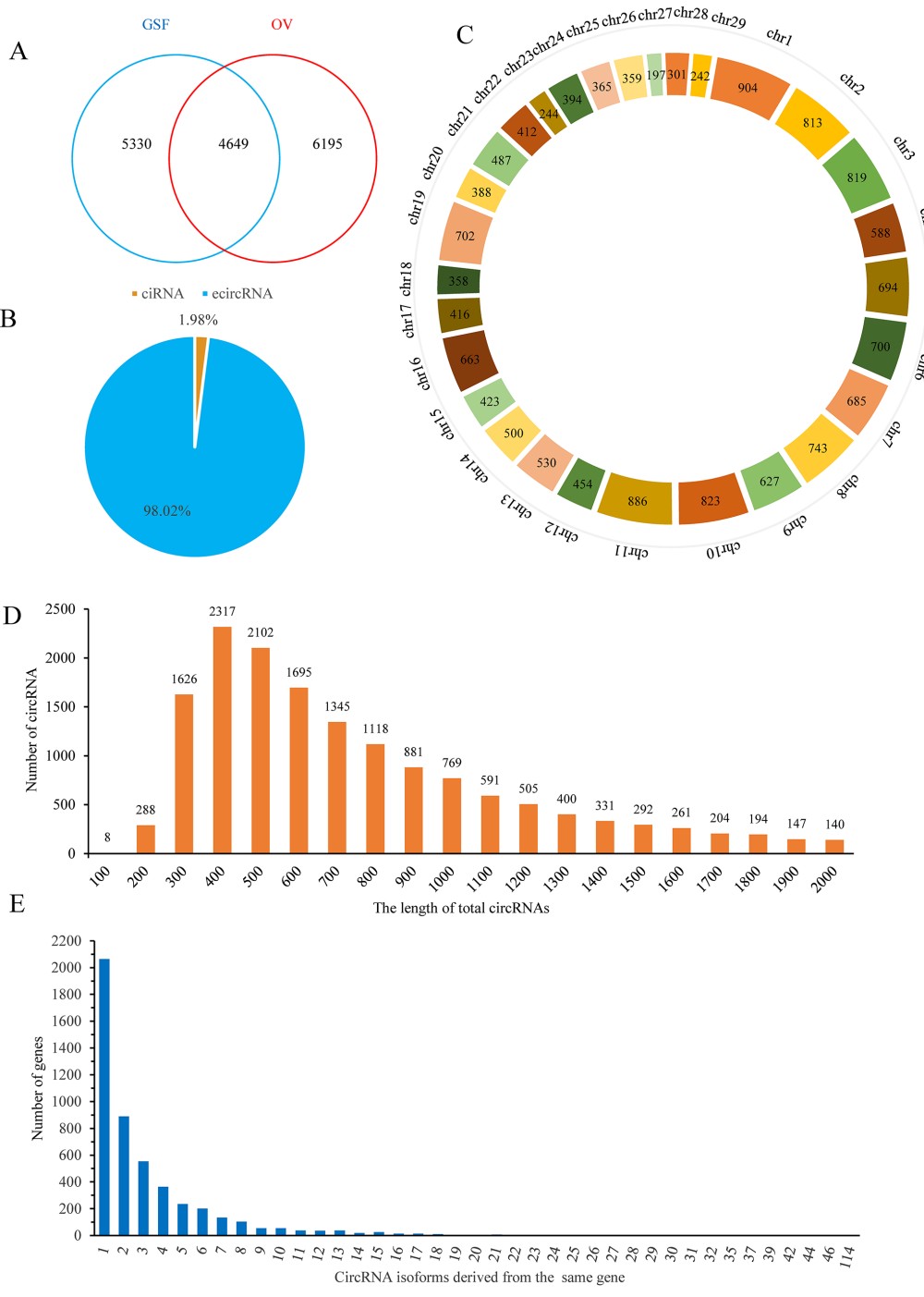

**Figure 1 Properties of circRNAs in ORFV-infected and uninfected GSF samples.** (A) Venn diagram showing the number of circRNAs either shared between or uniquely expressed in the GSF samples and ORFV-infected samples. (B) Pie chart showing the prevalence of the different circRNA types. ciRNAs from introns are represented by the orange section, while the blue region indicates ecircRNAs from exons. (C) Distribution of circRNAs among the chromosomes of the goat reference genome. (D) Lengths of the circRNAs from the GSF and ORFV-infected samples. (E) Number of circRNA isoforms derived from the same gene.

## Differential expression analysis of circRNAs, mRNAs and miRNAs

Based on the filtering criteria in materials and methods, a total of 151 circRNAs were identified as differentially expressed in ORFV-infected samples compared with GSF samples. Of these, 59 circRNAs were up-regulated while 92 circRNAs were down-regulated (Figs. 2A, 2B, Table S4). There were 341 differentially expressed mRNAs with 187 up-regulated mRNAs and 154 down-regulated mRNAs. The significant differences in transcript expression between OV samples and GSF samples was presented by clustered heatmap and volcano plot (Figs. 2D, 2E, Table S5). As for miRNA, 23 miRNAs were up-regulated while 33 miRNAs were down-regulated. Among the 56 differentially expressed miRNAs, 26 miRNAs were goat-derived in miRBase 21.0 and seven miRNAs (PC-3p-8215_174, PC-5p-406_14064, PC-5p-2253_1210, PC-5p-5127_361, PC-3p-10316_124, PC-3p-19472_48, PC-3p-4306_468) were first reported (Figs. 2A, 2C, Table S6).

## GO and KEGG analysis for host genes of differentially expressed circRNAs

Due to the fact that the biological functions of circRNAs may be associated with their corresponding parental transcripts, GO and KEGG analyses for host genes of dysregulated circRNAs were conducted in the study. The top 20 GO terms significantly enriched ($p \leq 0.05$) in molecular function, cellular component, and biological process are presented in Fig. 3A. The top six enriched GO terms in cellular component were proteinaceous extracellular matrix, intercalated disc, voltage-gated sodium channel complex, plasma membrane, T-tubule and extracellular region. The top six enriched GO terms in molecular function were growth factor binding, voltage-gated sodium channel activity, neuropilin binding, semaphorin receptor binding, chemorepellent activity and integrin binding. Moreover, regulation of inflammatory response, epithelial structure maintenance, negative regulation of insulin secretion, positive regulation of cell migration, positive regulation of ubiquitin-protein transferase activity, regulation of ion transmembrane transport were significantly enriched in the biological process subgroup (Fig. 3A, Table S7).

Next, we conducted KEGG enrichment analysis for circRNA-hosting genes. Nine pathways: Neuroactive ligand–receptor interaction, Tight junction, Rheumatoid arthritis, Transcriptional misregulation in cancers, Focal adhesion, Vascular smooth muscle contraction, Mismatch repair, Other types of O-glycan biosynthesis and Adherens junction were significantly enriched ($p \leq 0.05$). Furthermore, pathways such as Platelet activation, Fc gamma R-mediated phagocytosis, Endocytosis, Cytokine-cytokine receptor interaction, and TGF-beta signaling pathway were also enriched ($p > 0.05$). Fc gamma R-mediated phagocytosis and cytokine-cytokine receptor interaction were involved in host immune response to pathogen infection (Fig. 3B, Table S8).

## Integrated analysis of circRNAs-miRNAs-mRNAs

CircRNAs could serve as miRNA sponges indirectly regulating gene expression. Therefore, we constructed ceRNA networks based on up-regulated circRNAs, down-regulated miRNAs and up-regulated genes or down-regulated circRNAs, up-regulated miRNAs and down-regulated genes respectively to explore the biological functions of circRNAs during ORFV

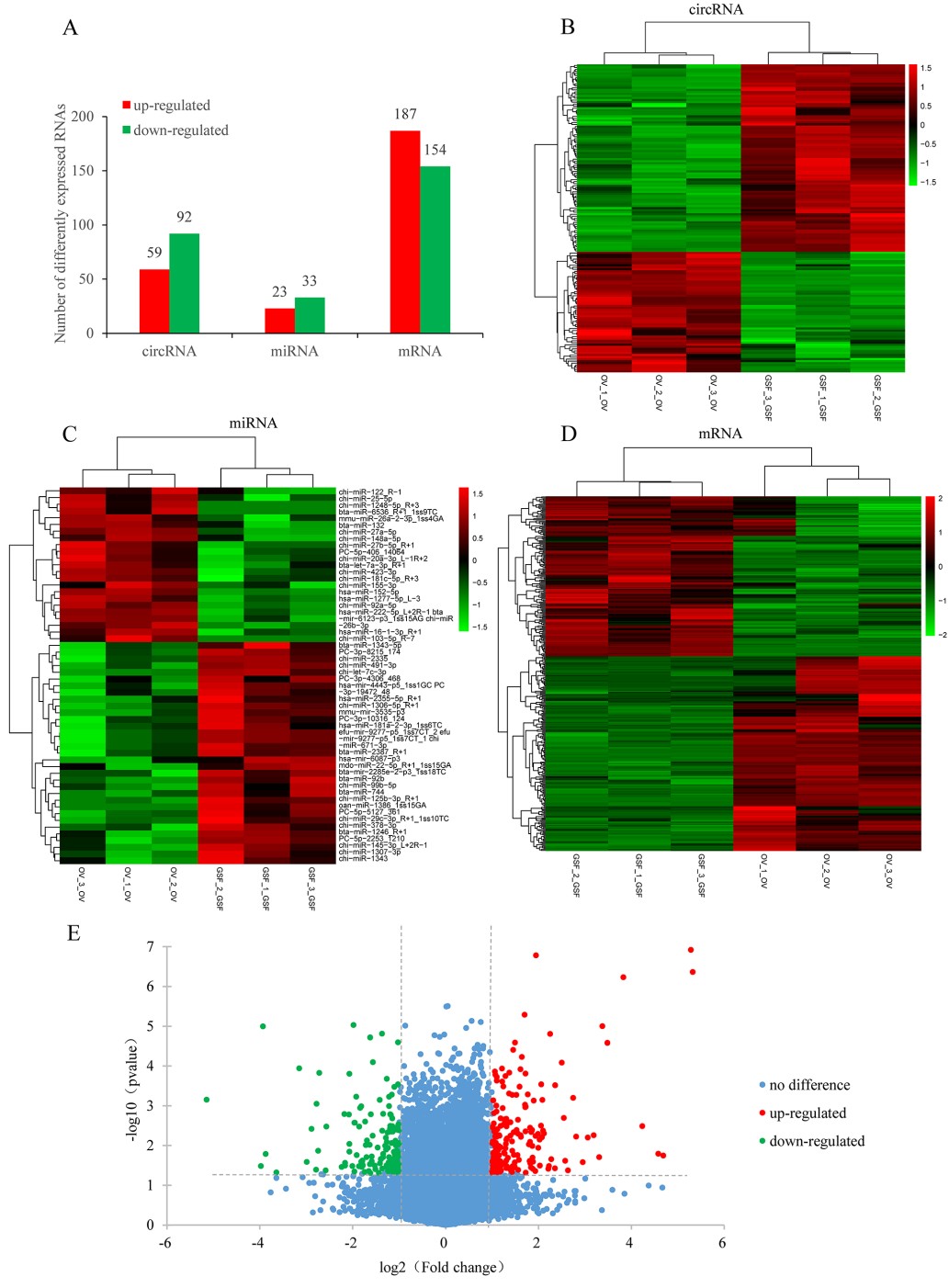

**Figure 2** **Differentially expressed circRNAs, mRNAs, miRNAs in ORFV-infected samples compared with GSF samples.** (A) Bar charts showing the number of differentially expressed circRNAs, mRNAs, miRNAs, respectively. Red bars, up-regulated RNAs; green bars, down-regulated RNAs. (B) Heatmap of differentially expressed circRNAs. (C) Heatmap of differentially expressed miRNAs. (D) Heatmap of differentially expressed mRNAs. Color from green to red; the deeper the color, the higher the expression. (E) Volcano plot of differentially expressed mRNAs. Vertical lines correspond to 2-fold changes in up-regulation and down-regulation. Horizontal line represents *p* value 0.05. Red points refer to up-regulated mRNAs; Green points refer to down-regulated mRNAs; Blue points refer to mRNAs with no significant difference.

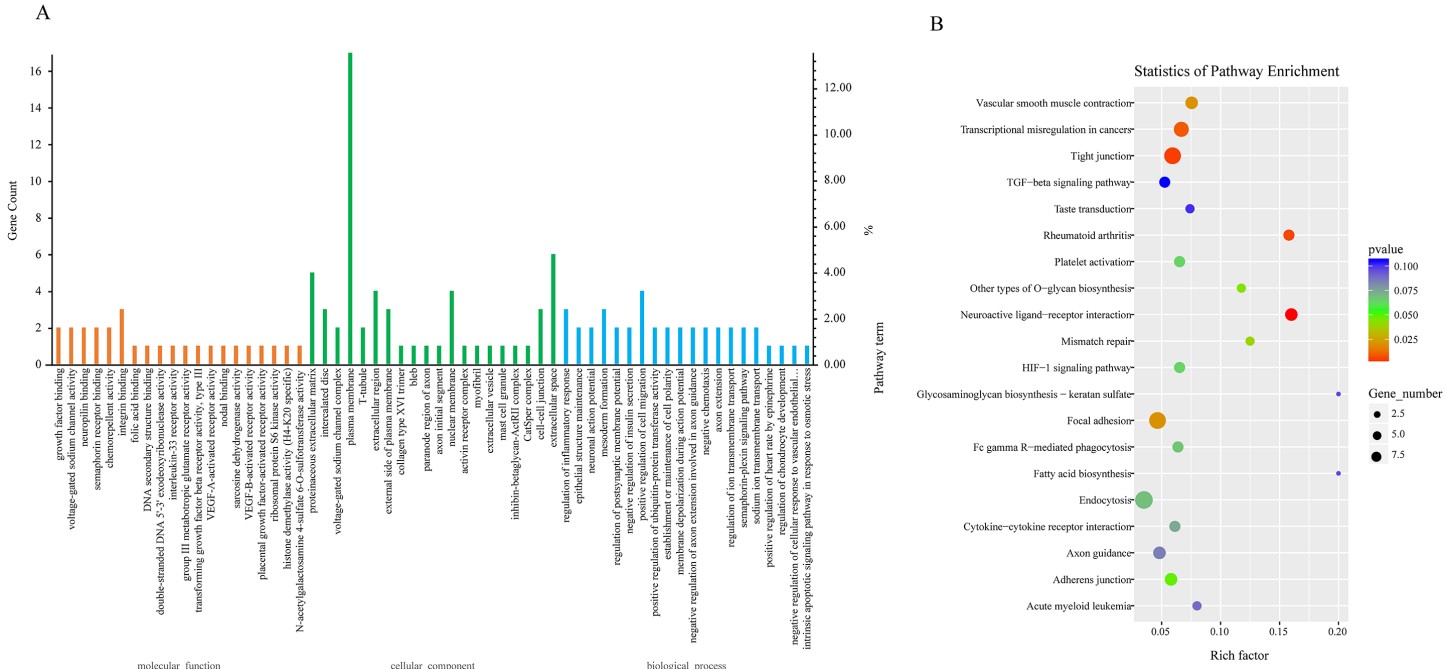

**Figure 3    GO and KEGG analyses for host genes of differentially expressed circRNAs.** (A) GO enrichment analysis for host genes of differentially expressed circRNAs. Top 20 GO terms ($P \leq 0.05$) in biological processes, cellular components and molecular functions are presented. Left $Y$-axis represents the absolute gene counts enriched in the term; right $Y$-axis represents % of the genes enriched in the GO term compared to the total number of genes enriched in top 20 GO terms in three ontologies. (B) KEGG pathway enrichment analysis for host genes of differentially expressed circRNAs. $Y$-axis represents pathways; $X$-axis represents rich factor; (rich factor equals the ratio between the host genes of differentially expressed circRNAs and all annotated genes enriched in the pathway); The color and size of each bubble represent enrichment significance and the number of genes enriched in a pathway, respectively.

infection (Fig. 4). Interestingly, we found that chi-miR-103-5p_R-7, chi-miR-26b-3p, chi-miR-92a-5p, and chi-miR-122-R_1 could bind at least four circRNAs. Novel miRNA PC_3p-10316_124 was shared by both circRNA-186 and circRNA-10457 while chi-miR-2335 and circRNA8386 showed unique binding. The validated circRNA1684 could function as sponge of chi-miR-92a-5p indirectly down-regulating eleven genes and the validated circRNA-3127 could bind chi-miR-103-5P down-regulating eight genes.

## Validation of miRNAs and circRNAs

The qRT-PCR results showed all six novel miRNAs could be specifically amplified. The expression levels of PC-3p-4306_468, PC-5p-5127_361 and PC-3p-10316_124 were consistent with the results of small RNA sequencing while the remaining three miRNAs did not show significant differential expression (Fig. S1).

The relative expression levels of nine circRNAs were validated by qRT-PCR using specific divergent primers. The results indicated that circRNA1001, circRNA1684 and circRN3127 were down-regulated while circRNA7880 was up-regulated, which were consistent with the RNA-seq data (Fig. 5A). As expected, the results of the agarose gel electrophoresis demonstrated that divergent primers could only amplify circRNAs from cDNA samples, while the convergent primers amplified products from both gDNA and cDNA. (Fig. 5B).

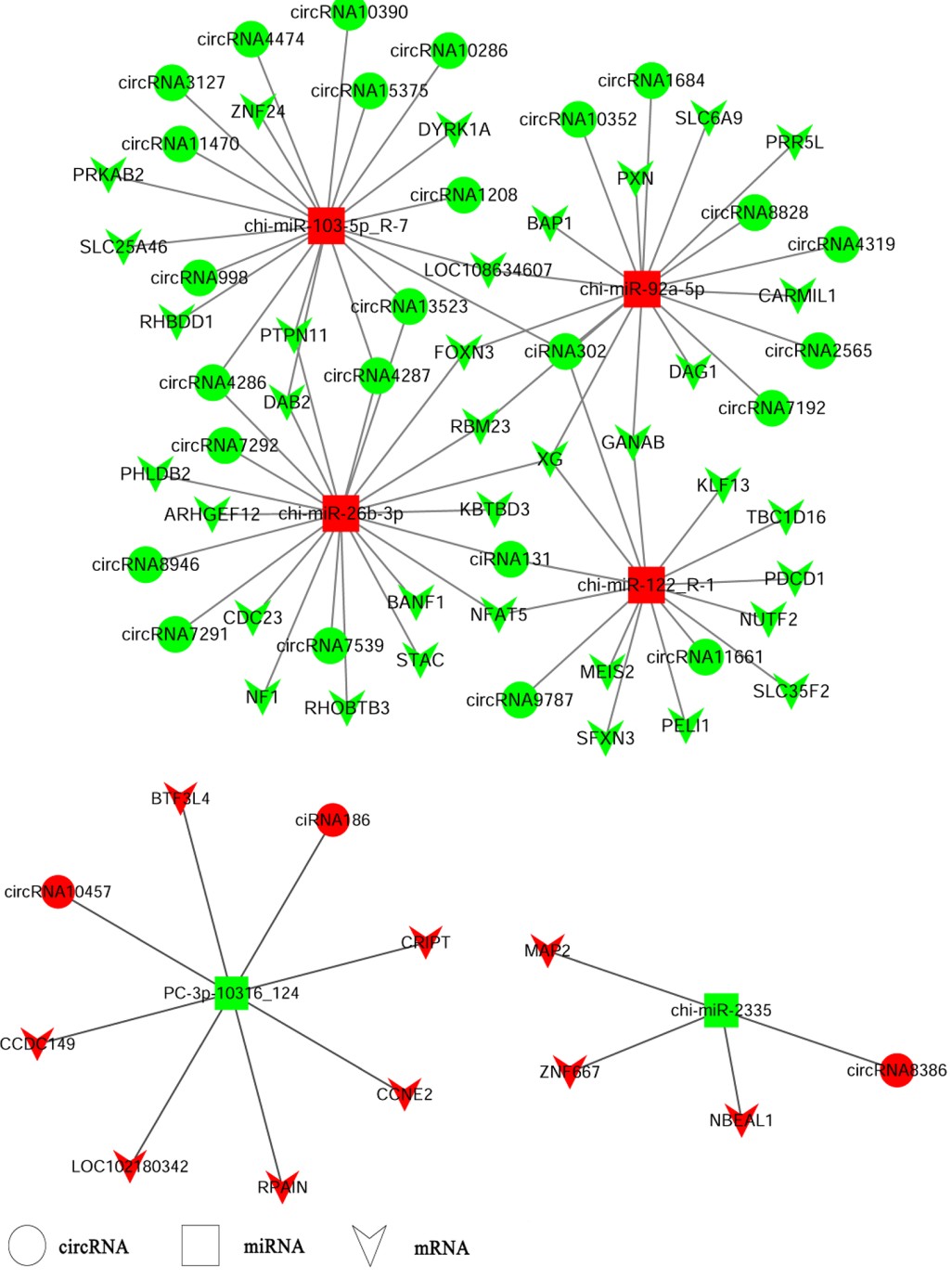

**Figure 4  A ceRNA network.** A ceRNA network based on up-regulated circRNAs, down-regulated miR-NAs and up-regulated genes or down-regulated circRNAs, up-regulated miRNAs and down-regulated genes. Red and green represent up- and down-regulation, respectively. The circle, rectangle and arrow represent circRNAs, miRNAs, and mRNAs, respectively.

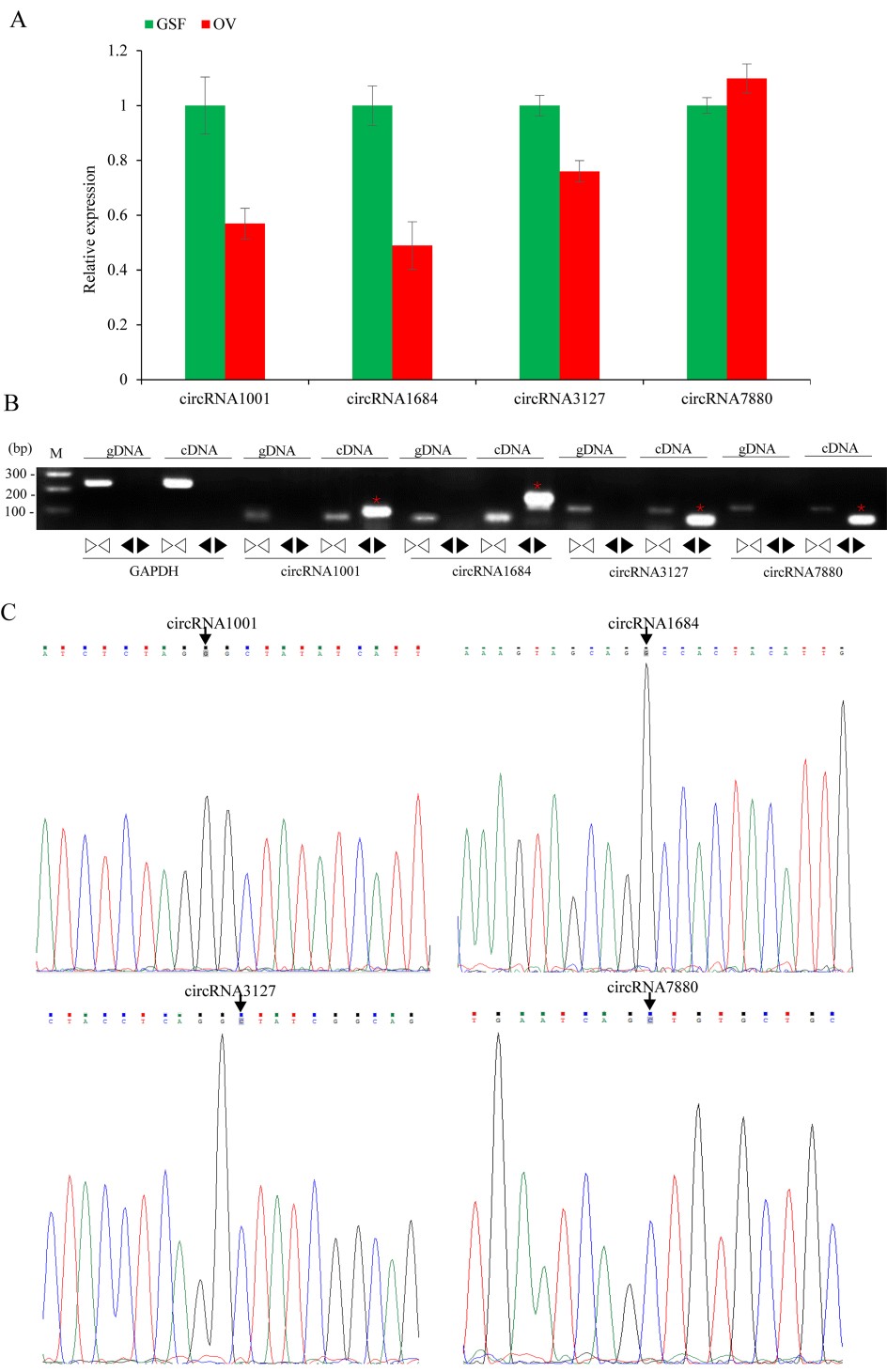

**Figure 5 Validation of circRNAs by qRT-PCR and Sanger sequencing.** (A) Validation of differentially expressed circRNAs using qRT-PCR. Data from qRT-PCR assays are the means of three independent replicates, with error bars representing SD. (B) The PCR amplification by divergent and convergent primers in genomic DNA and cDNA samples. The red asterisks represent the band of circRNAs from cDNA samples. The linear GADPH gene serves as an internal control. (C) Sanger sequencing confirmation of back-splicing junctions of circRNAs. Back-splicing sites are indicated with black arrows.

Back-splicing junctions of circRNAs from cDNA samples were further validated by Sanger sequencing (Fig. 5C).

## DISCUSSION

Transcriptional profiling is a powerful tool for researching the host-virus interactions during infection. Recently, researchers performed RNA-seq of sheep oral mucosa in response to Orf virus infection (*Jia et al., 2017*). They found that multiple differentially expressed genes were enriched in GO terms such as immune response, inflammatory response and apoptosis, indicating that the host could defend virus invasion through immune response and induce cell apoptosis to block viral proliferation. Another study reported alterations of transcriptional profiles in human foreskin fibroblast cells following ORFV infection (*Chen et al., 2017*). A variety of genes involved in immune response, apoptosis, cell cycle, etc were differentially expressed. These studies provided new insights into the mechanisms of infection by orf virus. Accumulating evidence indicated that circRNAs, a new type of non-coding RNAs, played key roles in diverse diseases (*Li et al., 2015*; *Tian et al., 2017*). *Li et al. (2015)* reported that circular RNA ITCH had inhibitory effect on ESCC by suppressing the Wnt/β-catenin pathway. Tian et al. revealed that hsa_circ_0043256 could inhibit cell proliferation and induce apoptosis by acting as a miR-1252 sponge. Although the mechanisms underlying ORFV-host interactions were thoroughly investigated, whether circRNAs were involved in the interactions between ORFV and its host yet remained unknown. Hence, in the current study, we conducted deep circRNA sequencing and small RNA sequencing to identify differentially expressed circRNAs, miRNAs and mRNAs and expected to find the potential circRNA–miRNA–mRNA network existing in interactions between ORFV and its host.

After circRNA sequencing, R package-psych was used to perform PCA (Principle Component Analysis) statistics investigating the correlations coefficient among the three virus-infected samples and the three mock samples. According to the values of each sample in the first principal component (PC1) and the second principal component (PC2), a two-dimensional coordinate map is made (Fig. S2). The results indicated that the three mock samples were close to each other and they were clearly distinct from the three virus-infected samples. There existed a good similarity among samples in the same group and the obvious distinction between the virus-infected samples and the mock samples. Based on the existing criteria for identifying candidate circRNAs, approximately ten thousand circRNAs were detected in uninfected and ORFV-infected GSF cells. Previous studies have also demonstrated that circRNAs are abundant in mammalian cells (*Guo et al., 2014*; *Salzman et al., 2012*). However, if we only include the circRNAs detected in at least two of the three replicates in uninfected or ORFV-infected GSF samples, the number of candidate circRNAs decreased to a large extent. There were only 2,935 circRNAs detected in uninfected GSF samples while 2,359 circRNAs were detected in OV samples (Tables S9, S10). This would significantly reduce false positives of candidate circRNAs identified in GSF samples or OV samples. Furthermore, we found that >98% of circRNAs were ecirRNAs, which differs from Hu's finding in which 86% of circRNAs were ciRNAs while only 13% were ecircRNAs and

1% were exon-intron circRNAs derived from the pre-ovulatory ovarian follicles of goats (*Tao et al., 2017*). These discrepancies further demonstrate that the expression patterns of circRNAs are tissue specific and cell specific.

ORFV infection influenced the circRNA expression profile of the host cells. Compared with the GSF samples, 151 differentially-expressed circRNAs derived from 90 parental genes were identified. Our findings showed that a single gene locus could produce one, two, or several circRNAs through alternative splicing (Table S4). Diverse circRNA isoforms derived from the same cognate linear gene were differentially expressed in ORFV-infected GSF cells, indicating that their parental genes played significant roles in regulating the temporal expression of circRNAs. Fang et al. (*Fang et al., 2018*) recently reported that circ-Ccnb1 derived from its parental gene CCNB1, a regulator of cell mitosis, had an inhibitory effect on breast cancer cell proliferation and survival. The authors suggested that the biological functions of circRNAs might be closely associated with its parental gene. Next, we performed GO and KEGG analyses for the cognate linear isoforms of the differentially expressed circRNAs to explore the biological functions of circRNAs in response to ORFV infection. In the biological process oncology, regulation of inflammatory response, negative regulation of insulin secretion, positive regulation of cell migration, positive regulation of ubiquitin-protein transferase activity, regulation of ion transmembrane transport were significantly enriched with $p \leq 0.05$ (Fig. 3A, Table S7). CircRNA12709, circRNA14794 and circRNA14795, enriched in GO term "regulation of inflammatory response", were different isoforms of their parental gene TNIP (TNFAIP3-interacting protein (1) which were up-regulated in OV samples compared to GSF samples during ORFV infection. TNFAIP3 (tumor necrosis factor $\alpha$-induced protein (3) also called A20 encoded a ubiquitin-editing protein which was an inhibitor of NF-κB. TNIP1 was shown to play a role in NF-κB inhibition by interacting with A20 (*Aya et al., 2010*). Pathways such as Tight junction, Rheumatoid arthritis, Transcriptional misregulation in cancers, Focal adhesion, Vascular smooth muscle contraction, Mismatch repair and other types of O-glycan biosynthesis were significantly enriched ($p \leq 0.05$). These findings indicated that differential expression of circRNAs may be involved in many biological processes and cellular response to ORFV infection, providing us some valuable clues about the functions of circRNAs. Given that circRNAs could function as miRNA sponges regulating gene expression. We constructed a ceRNA network to explore the potential functions of differentially expressed circRNAs during ORFV infection. For example, circRNA302, circRNA1684, circRNA2565, circRNA4319, circRNA7192, circRNA8828 and circRNA10352 were predicted to sponge chi-miR-92a-5p. Also, circRNA131, circRNA302, circRNA9787, circRNA11661 potentially bound chi-miR-122. These results revealed that a potential ceRNA regulatory network existed in the host-ORFV interaction, although the exact regulatory mechanism requires further investigation.

## CONCLUSION

In conclusion, we identified 9,979 and 10,844 circRNAs in GSF cells before and after ORFV infection. A total of 151 circRNAs (59 circRNAs up-regulated and 92 circRNAs down-regulated) 341 mRNAs, and 56 miRNAs were differentially expressed following ORFV

infection. Four circRNAs: circRNA1001, circRNA1684, circRNA3127 and circRNA7880 were validated by qRT-PCR and Sanger sequencing. Host genes of differentially expressed circRNAs were significantly enriched in many biological processes including regulation of inflammatory response, positive regulation of cell migration, and regulation of ion transmembrane transport. A potential circRNA-miRNA-mRNA regulatory network exists during ORFV infection. Our study is the first to present the expression profiles of circRNAs in GSF cells in response to ORFV infection and may provide new insights into the mechanism underlying ORFV pathogenesis.

## ACKNOWLEDGEMENTS

We thank Tamsin Sheen, PhD, from Liwen Bianji, Edanz Editing China (http://www.liwenbianji.cn/ac), for editing the English text of a draft of this manuscript.

### Funding

This work was financially supported by the Key Research and Development Program of China (No. 2018YFD0501904), the Key Science and Technology Project of Hainan (ZDKJ2016017-01), Special Funding Projects for Local Science and Technology Development Guided by the Central Committee (ZY2017HN07), the China Agriculture Research System (CARS-38). The funders had no role in study design, data collection and analysis, decision to publish, or preparation of the manuscript.

### Grant Disclosures

The following grant information was disclosed by the authors:
Key Research and Development Program of China: 2018YFD0501904.
Key Science and Technology Project of Hainan: ZDKJ2016017-01.
Local Science and Technology Development Guided by the Central Committee: ZY2017HN07.
China Agriculture Research System: CARS-38.

### Competing Interests

The authors declare there are no competing interests.

### Author Contributions

- Feng Pang conceived and designed the experiments, performed the experiments, analyzed the data, prepared figures and/or tables, authored or reviewed drafts of the paper.
- Mengmeng Zhang, Xiaojian Yang, Guohua Li and Shu Zhu performed the experiments.
- Xin Nie, Ruiyong Cao, Xiaohong Yang and Zhenxing Zhang contributed reagents/materials/analysis tools.
- Haifeng Huang, Baobao Li and Chengqiang Wang prepared figures and/or tables.
- Li Du conceived and designed the experiments, approved the final draft.
- Fengyang Wang conceived and designed the experiments, authored or reviewed drafts of the paper, approved the final draft.

## Data Availability

GenBank: GSE121725 and GSE121726.

## Supplemental Information

Supplemental information for this article can be found online at http://dx.doi.org/10.7717/peerj.6267#supplemental-information.

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
