# Peer review of "Genome-wide analysis of circular RNAs in goat skin fibroblast cells in response to Orf virus infection"

_PeerJ, doi:10.7717/peerj.6267_

## Round 0.1 · original submission · Major Revisions

The reviewers raised some serious concerns, which I encourage you to read through and consider carefully. In general, more detail is required in the methods section. Additionally, both reviewers expressed concerns about experimental design, including the differential expression and network analyses. The reviewers' concerns would have to be addressed fully in a revised manuscript or the manuscript would have to be rewritten to include a subportion of the data with more accompanying details.

Reviewer 1 ·

Basic reporting

The manuscript is well written and organized.

Experimental design

So far as I can tell, the appropriate experimental design was used.

Validity of the findings

The miRNA results and network analyses are suspect. Please see general comments.

Additional comments

The manuscript by Pang and colleagues reports on the analysis of RNAs that are differentially expressed in cell cultures infected with Orf virus. They find numerous novel circRNAs in their dataset, show that the majority are derived from exons and go on to propose miRNA-mRNA-circRNA pathways using differentially expressed RNAs revealed in their study. circRNAs are an exciting new class of ncRNAs who’s functions are an active area of research and information that adds to this field is of value/interest. The manuscript is well written and easy to follow, but there are some serious concerns that the authors need to address before publication. The work that focused on identifying circRNAs was, to my knowledge, done appropriately, and the validation of several circRNAs experimentally makes this part compelling. The remaining work on miRNAs and the network analysis is, however, very concerning. The majority of differentially expressed miRNAs are not from the species under analysis! The downstream networks, making use of miRNAs that come from humans, mice, etc., are likely not real. I think separating the circRNA work and presenting it alone would be one route for salvaging this manuscript. To keep the miRNA analysis and network analysis, they’d have to go back to the beginning and do extensive experimental validations to propose any networks.

Comments:

Although the focus of this study was on the effects of Orf infection on host cell RNAs, it would be interesting to also analyze the transcripts coming from the Orf virus. Indeed, when they performed their initial alignment, it would have been a good idea to align to the Orf virus genome simultaneously to the goat reference genome. Without redoing the whole alignment, it would be interesting to remap reads that did not align to goat to the Orf virus genome and use their pipeline to deduce any potential viral circRNAs. This would definitely strengthen this current manuscript.

Lines 182-3, I have my doubts that the novel miRNAs reported here are real. Looking at Table 4, it is concerning that of the 26 differentially expressed miRNAs, only 7 are from goat; the others are from a variety of other species (e.g. humans, mice), as well as these putative novel miRNAs. This suggests that these might be sequencing artifacts. If they could be validated via stem-loop PCR and if there was evidence for their conservation in other species (e.g. their precursor hairpins), this could raise confidence in their existence.

Could the authors explain why the minority of their miRNAs were not from goat? This is very concerning and it raises questions about all subsequent analyses. The authors should consider reanalyzing the miRNA-Seq data using a different pipeline. Additionally, validating the seven goat miRNAs using stem-loop RT-qPCR would help raise confidence in their putative expression change.

Most of the miRNA nodes are for miRNAs from other species, or for the putative novel miRNAs. The presented circRNA-mRNA-miRNA networks are likely spurious. Are the networks involving goat miRNAs even plausible? I looked at the circRNA184 to chi-miR-92a-5p network and they seem to make sense: circRNA down, miRNA up, genes down…but how do the other nodes look?

Lines 208-16 are confusing. The Figure caption indicates that Fig. 4 is for all diff. expressed RNAs, but this section is included in the integrated analysis section. Were the GO and KEGG analyses performed on only the genes involved in the network shown in Fig. 3?

·

Basic reporting

This paper is the first to report the deep-sequencing analysis of goat skin fibroblasts (GSFs) infected with Orf virus (OV), including mRNA, micro-RNA and circular RNA analysis. The paper is very well-written and provides a useful insight into host-pathogen interactions. However significantly more detail is needed in the Methods and particular figure legends. The authors also need to clarify certain aspects (for example, why particular circRNAs were chosen for qRT-PCR confirmation over others) to make it easier to read. The Discussion needs to be expanded, specifically with comparison to previous Orf virus studies.

Minor alterations:
- Throughout the text, the authors should clarify whether "upregulated" actually means "significantly overexpressed in virus-infected cells". The use of vague terms such as "up" and "down" is not sufficiently clear.
- Line 115: "normlize" > normalize
- Line 186: "Due to biological functions..." > "Due to the fact that the biological functions..."
- Line 200: "an joint" > "a joint"
- Figures 2A, 4A and 5A need Y axis titles

Experimental design

- The experimental design describes three plates of infected and three non-infected (mock) plates as controls. However following RNA sequencing, all transcripts were combined (line 103). Why were all samples combined? Differential expression analysis is only possible when duplicate (or higher) samples are analysed in parallel, against control samples. This needs clarifying in the text. At the same time, it is insufficient to simply state "perl scripts were used" without including further details as to what functions those scripts performed. Finally, when reads were mapped to the reference genome (line 100) did the authors map to mRNA only (coding regions) or to non-coding as well?

- Line 121: Why was an absolute log(2) fold change threshold of 0.58 used for the miRNA analysis?

- Volcano plots (or similar) should be included for displaying the mRNA differential expression results.

- Whilst gene ontology analysis is widely used, there is not a singular standard approach. The authors need to explain in more detail how these analyses were conducted (lines 123 onwards). What dataset was used as the background? How was the data corrected?

- The legends for Figure 1, 2 and 3 need significant expansion. They should explain the figure without the reader having to refer to the main text.

- The heatmaps (Figure 2) are useful for quality control checks. Another quality control measure which should be included is a biological coefficient of variation figure, to confirm the similarities between the three virus-infected samples and disparities between these and the three mock samples. An example is Figure 2 in He et al., 2017, which is already referenced by the authors.

Validity of the findings

- No evidence is provided that the infection was successful. I would expect the number of reads mapping to the viral genome, for each infected and mock sample, to be included in the analysis (Table 2). Additionally, why is this table only included for the circRNA libraries and not the miRNA (small RNA) libraries, which according to the Methods were processed separately?

- I strongly feel that the high number of circRNAs identified should be interpreted with caution (initially described at line 162). The authors should include additional measures to avoid false positives, such as those described in He et al., 2017. For example, I would suggest only including those circRNAs which were identified in at least two of the three replicates. Given the impressive depth of coverage this should be feasible. If the authors strongly feel that this stringency is not necessary, they should explain why.

- Figures 3 and 6 should be coloured according to which circRNAs, miRNAs and mRNAs were significantly overexpressed in either virus-infected or mock samples. This would significantly aid in interpreting the data.

- Figures 4 and S1: The GO analyses should highlight which terms arose from which dataset (either infected or mock), rather than listing them in a single histogram. It should be stated explicitly in the text that very few of the KEGG pathways are significantly enriched, as indicated by the p values. Additionally, the phrase "gene_number" is unclear. Is this an absolute gene count, in which case this is incorrectly performed? These analyses should be based upon a ratio of the number of differentially expressed genes, with that particular GO/KEGG annotation, compared to a predefined background (or example, the total number of differentially expressed genes in that sample).

- Why were those particular circRNAs chosen for confirmation by qRT-PCR? This needs to be explained further in the text. Additionally, there is no evidence that the qRT-PCRs are amplifying circRNA only, and not contaminating gDNA or linear mRNA transcripts. A figure similar to Figure 2 in Shi et al., 2017 is necessary (this is already referenced by the authors).

- Whilst the authors reference two previous studies of OV transcriptomics in the Introduction (Chen et al., 2017 and Huaijie et al., 2017), these are not discussed at all in the Discussion. The current dataset should be compared to these published works.

Additional comments

This dataset looks very interesting, however as it currently stands the analysis is not stringent enough (or it needs further explanation) to allow accurate inferences to be made. Additionally, the data needs to be presented more clearly. I feel that once these aspects are altered it will prove to be of significant interest to the virology community.

---

## Round 0.2 · Minor Revisions

Upon review of your revised manuscript, Reviewer 1 has asked for major revisions and Reviewer 2 has asked for minor revisions. Therefore, my decision is 'minor revisions.' In the case of both reviewers, the majority of revisions requested deal with more rigorous validation of the findings in this study. Additionally, please take a second look at Supplementary Table 1 as per Reviewer 2's comment.

At this point I highly encourage you to see what you can do to address these two reviewers' comments, which will require looking back at Reviewer 1's comments from the first round of reviews, as Reviewer 1 feels his/her comments were not adequately addressed.

Reviewer 1 ·

Basic reporting

Well constructed manuscript revision.

Experimental design

Experimental design is adequate.

Validity of the findings

Needs further validation.

Additional comments

I mentioned in the previous review that I think they should validate the diff. expression of their novel miRNAs at least using RT-qPCR. This was not done. In general, the authors response to my suggestions (and of the other reviewer) were basically to restate what was reported in their methods section.

·

Basic reporting

No comment

Experimental design

No comment

Validity of the findings

The manuscript is greatly improved and the Figures are now clearer and easier to interpret. Most of my initial concerns have been addressed to a satisfactory level. The online title of Supplementary Table 1 (as listed on the manuscript page within peerJ) is currently incorrect, however the data within it matches the description within the manuscript text.

There are only two remaining concerns; the text should be altered to include these caveats to the conclusions:

1) The amendment at line 263 regarding false positives is inadequate. At either line 108, or within the Discussion (line 267), the authors should explicitly state how many circRNAs were identified in at least two of the three replicates. If none of the altered circRNAs were found in more than one sample, this should be clearly stated. This would aid in the reader in judging whether a circRNA is reproducibly up- or down-regulated following virus infection, or whether some of the reads were “noise”. The circRNAs which appeared in multiple samples would be of significant interest and should be listed (either as a table or within the text).

2) The qRT-PCR still does not have the essential controls. The text should be amended to state that neither mRNA nor reverse-transcriptase-free reactions were included (I suggest at line 237) and therefore the products may be formed from either linear RNA or genomic contamination. It remains possible that the splicing junctions are actually SNPs in the cellular genome. I agree this is unlikely, and the PCR product sequencing looks convincing, but that is no reason not to include controls.

---

## Round 0.3 · Minor Revisions

I am satisfied with your responses to the reviewers' comments and I thank you for making the revisions to your manuscript. I have made tiny edits throughout the manuscript, mostly grammatical or to correct typos here and there. Please incorporate these tiny edits if you agree and then I think this manuscript will be in good shape.

---

## Round 0.4 · accepted · Accept

Thank you for doing that last round of editing.

#